# Peer review of "Comparison of inertial records during anticipatory postural adjustments obtained with devices of different masses"

_PeerJ, doi:10.7717/peerj.15627_

## Round 0.1 · original submission · Major Revisions

Dear authors, as you can see from the opinions of the 3 reviewers, different analyses were obtained from your article. I agree with the most critical reviewers that fundamental changes are needed for the article to be published. However, the topic is interesting, and the use of IMU's and accelerometers has high potential in clinical reality.

There are flaws in the writing and in the designing of the paper.
And, I would like to offer a round of opportunity for you to try to solve these problems. Please answer all the reviewers' remarks carefully, point by point, and indicate in the reply letter the location of the changes in the ms. Improve the consistency between the reasoning of the introduction, hypothesis, and discussion. There are some criticisms about the methods that should be carefully answered, especially about the analysis of the data coming from the IMU's.

best regards,
Leo

Reviewer 1 ·

Excellent Review

This review has been rated excellent by staff (in the top 15% of reviews)
EDITOR COMMENT
Indeed, the review was outstanding, mostly because the reviewer presented a deep analysis regarding the rationale in the introduction and discussion and regarding acquisition and processing methods. Also, the reviewer skillfully and analytically debates study's question on rational basis. More than asking for specific papers, the reviewer asked for some rationale thorough the paper, clever. Thanks for the constructive revision and careful analysis.

Basic reporting

I suggest the authors reduce the references used to quote simple events. In some cases, there are more than four quotes – e.g., Lines 61-62). It also occurs in several other places.

I recommend that the authors to highlight the main findings from table 1 – presented in the introduction section, and move it to the methods section (or remove it). It is not usual to present tables at the introduction.

Experimental design

At the end of the introduction, the authors present some issues regarding the studies on identifying APAs before step initiation but need to state the present study's purpose clearly. Please, state objectively what makes the present study differs from previous ones. I also suggest that the authors present robust information and rationale regarding how the weight of the accelerometer influences performance. I understand the vibration issues, but recently, most sensors have been very light. Is it a real problem or something revived from the past? It is further confusing as the present study does not incorporate any analysis using a “heavyweight” sensor. So, comparisons with other studies are somewhat difficult since many factors influence the results (e.g., speed, performer, etc.). In terms of practical design, it can only be responded to by measuring the same phenomenon simultaneously, which is not the case in the present study. I question why the study is relevant after others have reported “excellent” intra-session reliability. I see no purpose in assessing the reliability analysis twice on alternate days, two weeks apart. What does it reveal? It is likely to reveal subjects' variability rather than the accuracy or reliability of your measures.

To add to the relevant issues of the study, please, consider that accelerations were calculated from displacement (the second derivative). Are these measurements error-free?

The design applied needs to be revised. Comparing the performance between sessions requires the performers to execute the same movement, which is impossible. The outcomes include the variability of the participants (which varies over time) and not of the devices.

I don't know whether the number of participants was provided.

Is the FS a good instrument? I ask this because there is a certain threshold it is able to identify the start of the movement; however, the initiation has started well before the FS can detect it since it detects the instant the heel leaves the ground. Please remember that the initiation has long been in progress when it occurs. Furthermore, a description of the FS needs to be included. Is it an ON-OFF switch? If so, all arguments that it is inaccurate to identify gait initiation are true. I see no use of the FS in your analysis. What was it used for?

Stating that the syn was performed using MATLAB is challenging as several approaches can be used for such an approach. In addition, the quote about the vertical jump is also vague.

How the variables were defined when assessed via accelerometers. I am curious to know how the instant of the maximal amplitude and the time were defined. In addition, nothing is mentioned regarding the displacements were calculated using the accelerometer. Have the authors used the second integral? Or just the displacements were derived to calculate acceleration?

Correlations are inappropriate. Please, justify and explain why such an analysis applies. I'm afraid I have to disagree. Near-perfect correlations may still present considerable differences in the magnitude of the differences in your signals. The relative error is a more interesting measure.

Furthermore, correlations are likely to be significant when one uses a large sample. Please note that what you call “very high correlation” (0.86) explains about 73% of your variance. Thus, a large part is unexplained.

What do your green lines mean in Figure 4 (?). Note that several points fall out of your confidence lines. Am I wrong?

Again, nothing is mentioned or is present in your methods that entitle the authors to assess the issue regarding the weight of the devices.

The presentation of the actual values of your variables of interest is mandatory. In the present form, only the ICCs are indicated.

I confess that I got lost when comparisons between days were performed. I cannot identify where these assessments were shown.

To conclude, note that some measurements were borderline to 1.96 SD, which is critical regarding reliability analysis, as they are considerably far away from the mean.

Validity of the findings

The findings are limited in light of the aims. Note that the weight of the accelerometers was not assessed. Comparisons between days are unclear, while the definition of the variables is difficult to understand and, therefore, to support the findings.

Additional comments

Nothing to declare

Reviewer 2 ·

Basic reporting

The authors used a clear and unambiguous language. The figures and tables are relevant to the study. I suggest that Table 01 be inserted in the discussion and that, in Table 02, the authors delete the second phrase of the title. The asterist (*) was not shown in the results displayed in the table.

Experimental design

The aim of the research question was well defined and relevant. The methods were well described.
I suggest authors provide additional data about the sample (number of men and women; mean and standard deviation of height, age, weight and BMI). How was the sample size calculated? The supplementary document to the manuscript presents n=21, but in the methods n=18 (3 subjects were excluded?)

Validity of the findings

The research findings are interesting and discussed based on current references. The authors believe that the weight of the sensor (5.6 g) may have an impact on its reliability. Could the authors improve the discussion comparing with the results of the your previous study using Metamotion C (28.34 g) ? (Duarte et al., 2022; DOI 10.3390/s22218272)

·

Basic reporting

The English language should be improved to ensure that an international audience can clearly understand your text. Some examples where the language could be improved include lines 196-199; 237-238– the current phrasing makes comprehension difficult. Besides, several errors during the formatting are visible such as the lack of spaces and final dots.

The content structure is reasonable, but the main result contradicts the hypothesis raised by the authors. Therefore, I suggest a complete change of the rationale to be worth the justification for using lightweight sensors; otherwise, the text does not correspond to that objective.

Experimental design

The Experimental design has an important issue to be considered. The imposition of the right foot to analyze APA in gait initiation is unjustified since the method does not allow a natural movement from this protocol. It is already known that the non-preferential limb provides a greater lateral impulse on the ground during gait initiation (doi:10.1016/j.gaitpost.2011.01.008).

Validity of the findings

No comment.

Additional comments

The manuscript seems innovative since it used a new IMU sensor to detect APA variables during gait initiation. Furthermore, the creative standpoint refers to using the lightest sensor on the market (referred to by the authors), which might justify its use in the field compared to heavier ones (i.e., smartphones). However, the authors contradicted themselves throughout the text to justify their choice, which might compromise their findings:

1. The authors raise the question of the sensors' weight and the lack of this information from some studies published on gait analysis outputs. Nevertheless, they suggested that "adding mass loading" (line 82) might affect results by engaging the tremor of the hands as a movement to analyze; in parallel, they pointed out that "the weight of the sensors may contribute to increased signal noise" (line 90), again bringing the reader to believe that lighter sensors would be more suitable for this aim. After this rationale, it is revealed that only three previous studies had reported 'excellent' reliability intra-session with IMU sensors to measure APAs at gait initiation. Surprisingly, the two heaviest sensors (>100 g; Martinez-Mendez et al., 2011 and Moraes et al., 2022) reported better results (validation and reliability), whereas the third study cited by the authors (Mancini et al., 2016) presented controversial results using lightest sensors. Contemporaneously, the summary table registered by the authors revealed that more lightweight sensors did not show reliable results. Then, justifying this use from the lightest point of view of sensors falls into contradiction when the authors reported the opposite results from the literature.

2. Line 65: Why do you not use the term IMU (Inertial Measurement Units) to define these sensors?

3. Line 97: The authors seem repetitive in declaring the aims and, firstly, seeking to 'test and validate' the new sensor for measuring APAs during gait initiation. However, the same 'validation' returns in lines 101-102. All these pieces of information might be gathered as the main objective of the manuscript in one sentence.

4. Lines 103-105: The structure of the Introduction does not lead to this hypothesis since the authors reviewed previous studies that highlighted heavier sensors as better indicators to validate and rely on APA variables in gait initiation. In this case, the hypothesis is not reasonable.

5. Lines 109-111: Please provide the exact number for each gender plus their average age and height.

6. Line 155: Was the right foot of all subjects dominant? If it did not, why obligated them to start from this foot? Why did they not choose their feet based voluntarily, likely assuming a more natural behavior since it has been reported the influence of limb preference in gait initiation (doi:10.1016/j.gaitpost.2011.01.008)?

7. Lines 191-192: Please provide the graph number.

8. Lines 196-199; 237-238: The English language should be improved to ensure an international audience can understand your text clearly. Please reformulate this content.

9. Line 244: Please indicate and reinforce that it is the first research validating this sensor for that aim.

10. Line 255: Please insert the platform and walkway in a plural form.

11. It is missing space in lines: 256, 263, 273, 277, and 280.

12. Lines 262-263: Please indicate the references which worked with these body regions.

13. Lines 283-285: Again, the authors contradict themselves since they had highlighted an advantage of using lighter IMU sensors compared to heavy ones for calculating APAs, but concluded the opposite.

14. Line 297: It is missing a final dot.

15. Line 302 (Conclusion): The manuscript did not go in-depth to discuss the lack of association raised by the authors between 'lighter sensors' and a supposed 'better reliability' beyond the contradiction to sustain their hypothesis. The use of light sensors, as exposed in the text, did not justify itself as a tool for that scope.

16. Table 1. Please add a variable column specifying the variables calculated for each sensor from each study.

17. Table 2. There is no need to insert the legend citing the asterisk since the p-value did not show significance.

18. Figure 4. Please cite in the legend that the correlations and Bland-Altman plots represent each variable analyzed. Also, do not forget to mention the features of the Bland-Altman graph (i.e., the number of standard deviations, explain the differences between the variables for each axle).

---

## Round 0.2 · Minor Revisions

As you can see in the analyses done by the reviewers, the paper is interesting and you can face just minor points to reply. I encourage you to submit your revised manuscript with tracked changes to facilitate the review and submit a reply indicating by lines/pages the exact position of the alterations related to points replied.

best wishes,
Leo A. Peyré-Tartaruga

Reviewer 2 ·

Basic reporting

No comment

Experimental design

No comment

Validity of the findings

No comment

Additional comments

The authors made the requests and improved the quality of the manuscript. I have an additional comment about Table 01. At the end of the title of the table is the following sentence "No significance differences were found". However, there is a significant difference in the values presented in the APAamp variable (p-value 0.004*). I propose to alter the title and insert the legend citing the asterisk since the p-value show significance.

·

Basic reporting

Overall, the manuscript has significantly improved after a substantial review. Some little issues even persist, such as misunderstandings written in the text, but they are considered of minor relevance.

Experimental design

No comment.

Validity of the findings

No comment.

Additional comments

The authors have substantially improved the manuscript with the changes proposed. As a result, the text seems much more straightforward, which enables us a right-way reading.
Some minor issues were highlighted, such as:

Abstract (pages 51-53)
Please rephrase. The Bland Altman's test validates a new method, comparing it with a gold standard (kinematic measurements).

Page 113
It is missing a space.

Page 120
There is a (supposed) unnecessary parentheses.

Page 127
Please, denote it as a separate phrase. In this version seems that the right-footedness participants were excluded as well.

Page 141
There is an extra dot.

Page 145
It is missing a space.

Page 289
It is missing a final dot.

Page 293
It is missing a space.

Page 300
It is missing a space.

Page 301
Please review: 'All of the studies' or 'All studies.'

Pages 336-338
Please insert a reference for this statement.

Page 341
It is missing a final dot.

Table 1
Is there not a significant difference in the table?

Figure 4
Please, again, indicate what is meaning 'Difference' and 'Average' in the graph.

---

## Round 0.3 · accepted · Accept

Congrats on the acceptance of your work. The paper was improved, and the tables are clearer. I hope the process has improved your paper.